# A Survey of Performance Metrics for Spectrum Sensing and Spectrum Hole Geolocation for Wireless Spectrum Access

**DOI:** 10.3390/s25123770

**Published:** 2025-06-17

**Authors:** Dayan Adionel Guimarães

**Affiliations:** National Institute of Telecommunications (Inatel), Santa Rita do Sapucaí 37536-001, MG, Brazil; dayan@inatel.br; Tel.: +55-35-3471-9227

**Keywords:** cognitive radio, dynamic spectrum access, hypothesis testing, performance metrics, spectrum hole geolocation, spectrum sensing

## Abstract

This paper presents a survey of performance metrics applicable to spectrum sensing and spectrum hole geolocation within the context of dynamic spectrum access (DSA) in cognitive radio networks. While grounded in binary hypothesis testing, the review emphasizes metrics specialized for sensing reliability, interference risk, spatial accuracy, and network efficiency. The work also highlights trade-offs among metrics and provides guidelines for their practical application. A key contribution of this work is to provide researchers and practitioners with a comprehensive set of evaluation tools, extending well beyond the applicability of the conventional probabilities of detection and false alarm.

## 1. Introduction

Spectrum sensing [1,2,3,4,5,6,7,8,9,10,11] is widely acknowledged as a fundamental process for enabling dynamic spectrum access (DSA) [12]. It involves monitoring the radio-frequency (RF) spectrum to identify available frequency bands, allowing secondary users (SUs) to share the spectrum with primary users (PUs), thereby enhancing the efficiency of spectrum utilization.

An available frequency band is often referred to as a spectrum hole, or white-space. It is a segment of the RF spectrum temporarily unoccupied by any licensed user or PU within a defined geographic region and time interval. In essence, it represents the space and time information on frequencies available for unlicensed SUs to employ for communication purposes without risking interference with the incumbent PUs.

The notion of spectrum holes emerges from the dynamic nature of RF usage and the environmental characteristics that affect signal propagation. While specific portions of the spectrum are allocated to licensed users, these users may not be active across all allocated bands at every moment and location. Consequently, certain areas and time periods often have portions of the spectrum that remain unused or underutilized, giving opportunities for secondary users to access these frequencies for their own communication needs.

When a secondary terminal searches for a vacant band to enable DSA, the spatial alignment of the spectrum hole with the terminal’s location is essential. As long as the terminal remains in a fixed position, a detected spectrum hole is beneficial only if it corresponds to that precise location; otherwise, it has no utility. Misalignment between the detected spectrum hole and the terminal’s location can result in unintended interference with the primary network.

It is important to emphasize that the declaration of a spectrum hole depends not only on the activity status of the PU transmitter but also on the propagation characteristics within the coverage area. Even if a PU transmitter is active, certain regions within its coverage area may experience diminished signal strength or attenuation due to factors such as distance, terrain, and obstructions. Consequently, SUs might still be able to operate within these regions without causing harmful interference to the primary communication system, as the PU network may not reliably reach these areas. This creates potential access opportunities for SUs without disrupting authorized services.

The detection of a spectrum hole in time, frequency and space can be formulated as a statistical decision process. Generally, statistical decision theory offers a structured approach for making decisions under conditions of uncertainty, a methodology integral to various disciplines, including economics, engineering, and applied sciences, where decision-making relies on data with inherent variability.

Statistical decision theory is a foundational element of mathematical statistics, particularly within the domain of statistical inference. It provides a rigorous framework for formalizing decision-making under uncertainty by employing probabilistic and statistical models to evaluate the potential consequences associated with different courses of action. The modern formulation of this theory was introduced by [13], who conceptualized decisions as actions linked to their possible outcomes through a loss function. Significant advancements have since been made, notably by [14], who extended the Bayesian decision-theoretic approach, and by [15], who developed methods for comparing statistical experiments.

In the context of spectrum sensing, statistical decision theory is frequently employed in the design of test statistics that support the associated binary decision process. Although a range of performance metrics can be used to evaluate such a decision-making process, spectrum sensing research typically emphasizes the use of two principal metrics: the probability of detection, $P_{d}$, and the probability of false alarm, $P_{\mathrm{fa}}$. The former denotes the probability of correctly identifying an occupied channel, while the latter represents the probability of incorrectly identifying a vacant channel as occupied.

While these two metrics are widely used in the literature and are sufficient in many scenarios, other metrics can be employed to provide a broader evaluation of spectrum sensing performance. Motivated by this consideration, the present study reviews a large set of metrics commonly used in binary decision processes and explores their adaptation and applicability to the context of spectrum sensing. Moreover, the survey also addresses the important process of spectrum hole geolocation, which refers to the localization of vacant bands in the spatial domain. Metrics tailored to this process are also discussed herein.

### 1.1. Related Work

The foundational survey presented in [1] introduces the concept of dynamic spectrum access, discussing spectrum sensing techniques, spectrum management, and the architecture of cognitive radio networks. The survey by [2] categorizes spectrum sensing techniques, including energy detection, matched filtering, and cyclostationary feature detection, while also addressing cooperative sensing and challenges such as noise uncertainty and sensing time. Ref. [3] delves into the fundamental limits of spectrum sensing, exploring detection performance under various channel conditions and proposing solutions to key challenges. The survey paper [4] provides an in-depth analysis of cooperative spectrum sensing methods, discussing various cooperation strategies, their benefits, and associated challenges, while exploring the trade-offs between sensing performance and cooperation overhead.

Wideband spectrum sensing techniques, including sub-Nyquist sampling and compressive sensing, are discussed in [5], which addresses the challenges of high sampling rates and computational complexity. A comprehensive overview of spectrum sensing techniques, including energy detection, autocorrelation, Euclidean distance, wavelet, and matched filter-based methods, highlighting their advantages and limitations, is provided in [6].

The extensive review in [7] covers narrowband and wideband sensing techniques, including compressive sensing and machine learning approaches, and addresses challenges such as hardware limitations and spectrum mobility. An in-depth analysis of wideband spectrum sensing algorithms, emphasizing sub-Nyquist approaches and their applicability in cognitive radio networks, is presented in the survey [8]. The survey in [9] explores recent advancements in spectrum sensing, emphasizing full-duplex paradigms, machine learning enhancements, and applications in IoT and 5G systems, while also outlining future research challenges.

The tutorial paper [10] provides an in-depth examination of spectrum sensing methods, including energy detection, matched filtering, and cyclostationary feature detection, highlighting their theoretical foundations, practical applications, and performance metrics such as detection probability, false alarm rate, and the SNR wall. Finally, the survey in [11] examines both traditional and modern spectrum sensing techniques, including machine learning-based methods, discussing their applicability in 5G cognitive radio networks.

### 1.2. Contributions and Organization of the Paper

This survey reviews a large set of metrics commonly used in binary hypothesis testing and explores their adaptation and applicability to spectrum sensing. Moreover, the survey also addresses the process of spectrum hole geolocation, which refers to the localization of vacant bands in the spatial domain. Metrics tailored to this process are also discussed.

Despite the substantial contributions provided by the aforementioned surveys and tutorials, none address performance metrics in spectrum sensing with the comprehensive depth found in the present work. While previous references primarily emphasize techniques, methods, theoretical foundations, or implementation challenges, this work uniquely offers an extensive and structured analysis exclusively focused on spectrum sensing and spectrum hole geolocation performance metrics. This includes detailed discussions on detection probability, false alarm rates, ROC curves, geolocation accuracy, and computational efficiency, thereby establishing a robust framework essential for the precise evaluation and comparison of spectrum sensing methodologies across various application scenarios.

The remainder of the paper is organized as follows. Section 2 introduces the statistical basis for binary decision-making in spectrum sensing. Section 3 addresses performance metrics applied to spectrum sensing. Section 4 presents geolocation metrics. Section 5 provides numerical examples and interpretations. Section 6 concludes the paper and outlines future directions.

## 2. Statistical Foundations for Spectrum Sensing

This section addresses the basics of statistical detection theory [16,17,18,19] applied to the spectrum sensing context.

### 2.1. Binary Hypothesis Testing in Spectrum Sensing

Spectrum sensing is commonly formulated as a binary hypothesis testing problem [20,21,22]. A secondary user must decide whether a primary user signal is present or absent in a frequency band, based on a sequence of observed signal samples. The hypotheses are defined as $H_{0}$ (the spectrum is idle, i.e., the PU signal is absent), and $H_{1}$ (the spectrum is occupied, i.e., the PU signal is present).

Let *X* be a random variable representing the received signal observation, whose statistical distribution depends on the underlying hypothesis. Under $H_{0}$, $X∼f_{0}\left(x\right)$; under $H_{1}$, $X∼f_{1}\left(x\right)$. A decision rule δ(x) maps an observed value *x* to one of the hypotheses, that is,(1)$$\delta \left(x\right)=\left\{\begin{array}{cc} H_{0} & \mathrm{if}\;x\in R_{0},\\ H_{1} & \mathrm{if}\;x\in R_{1}, \end{array}\right.$$
where $R_{0}$ and $R_{1}$ are the decision regions. The reliability of this decision process is characterized by two types of statistical errors [23]: in Type I Error (false alarm), $H_{0}$ is true, but $\delta \left(X\right)=H_{1}$ with probability(2)$$\alpha =\mathrm{Pr}\left[\delta \left(X\right)=H_{1}|H_{0}\right]=\int_{R_{1}}f_{0}\left(x\right)\;dx.$$

In Type II Error (missed detection), $H_{1}$ is true, but $\delta \left(X\right)=H_{0}$ with probability(3)$$\beta =\mathrm{Pr}\left[\delta \left(X\right)=H_{0}|H_{1}\right]=\int_{R_{0}}f_{1}\left(x\right)\;dx.$$

The probability of missed detection, $P_{m}=\beta$, is the probability that the test fails to detect the presence of a PU signal when the spectrum is occupied.

The probability of detection is defined as $P_{d}=1-P_{m}=1-\beta$, and the probability of false alarm as $P_{\mathrm{fa}}=\alpha$. The trade-off between them plays a crucial role in DSA: reducing false alarms increases spectrum utilization, while minimizing miss detections helps avoid interference to licensed users [2].

The quantity 1−β is also sometimes referred to as the statistical power of a test. Formally, it is the probability of correctly rejecting $H_{0}$ when $H_{1}$ is true, that is, $\mathrm{Power}=1-\beta =\mathrm{Pr}[\delta \left(X\right)=H_{1}|H_{1}]$.

### 2.2. Neyman–Pearson Criterion

The Neyman–Pearson (NP) framework provides a basis for constructing optimal decision rules for hypothesis testing under constraints [18,20,21]. It establishes that, for a fixed significance level α, the most powerful test, i.e., the one that maximizes $P_{d}$, is based on the likelihood ratio [24](4)$$\Lambda \left(x\right)=\frac{f_{1}\left(x\right)}{f_{0}\left(x\right)}.$$

According to the NP lemma, the optimal decision rule is(5)$$\delta \left(x\right)=\left\{\begin{array}{cc} H_{1} & \mathrm{if}\;\Lambda (x)>\eta ,\\ H_{0} & \mathrm{otherwise}, \end{array}\right.$$
where η is a decision threshold (or detection threshold) chosen to ensure that $\mathrm{Pr}[\delta \left(X\right)=H_{1}|H_{0}]=\alpha$.

This rule underlies classical spectrum sensing strategies, particularly energy detection in additive white Gaussian noise (AWGN), where *X* corresponds to the energy of the received signal over a sensing window. In such cases, the PDFs $f_{0}\left(x\right)$ and $f_{1}\left(x\right)$ are derived from central and non-central chi-square distributions, respectively, and η is set to meet regulatory or design constraints on $P_{\mathrm{fa}}$.

### 2.3. Bayesian Approach to Spectrum Sensing

The Bayesian approach [14,25,26,27] to spectrum sensing incorporates prior probabilities about spectrum occupancy into the decision process. Let $\mathrm{Pr}(H_{0})$ and $\mathrm{Pr}(H_{1})$ denote the prior beliefs about the absence or presence of the PU signal in the sensed band. Given an observation *x*, Bayes’ theorem yields the posterior probabilities(6)$$\begin{array}{cc} \mathrm{Pr}(H_{0}|x) & =\frac{\mathrm{Pr}\left(H_{0}\right)f_{0}\left(x\right)}{\mathrm{Pr}\left(H_{0}\right)f_{0}\left(x\right)+\mathrm{Pr}\left(H_{1}\right)f_{1}\left(x\right)}, \end{array}$$(7)$$\begin{array}{cc} \mathrm{Pr}(H_{1}|x) & =\frac{\mathrm{Pr}\left(H_{1}\right)f_{1}\left(x\right)}{\mathrm{Pr}\left(H_{0}\right)f_{0}\left(x\right)+\mathrm{Pr}\left(H_{1}\right)f_{1}\left(x\right)}. \end{array}$$

The Bayesian decision rule selects the hypothesis with the higher posterior probability, which is equivalent to minimizing the expected loss under a specified cost function. With a symmetric 0,1 loss function, the decision reduces to(8)$$\frac{\mathrm{Pr}\left(H_{1}\right)f_{1}\left(x\right)}{\mathrm{Pr}\left(H_{0}\right)f_{0}\left(x\right)}>1,$$
which corresponds to a likelihood ratio test with an adjusted threshold incorporating prior information.

Bayesian detectors are especially suitable in cooperative sensing or spatially correlated environments, where historical data or models can be used to update priors dynamically. Although optimal in the Bayesian sense, these detectors require accurate prior knowledge, which may not always be available or easily estimable in practice.

## 3. Performance Metrics for Spectrum Sensing

In the context of a binary hypothesis test, several metrics are commonly used to evaluate performance of spectrum sensing [28]. These metrics provide insights into the accuracy, precision, and reliability of the test in distinguishing between the null and alternative hypotheses.

### 3.1. Confusion Matrix

A confusion matrix is a performance measurement tool used to evaluate classification models [29]. In the context of spectrum sensing, it assesses the effectiveness of a spectrum sensing algorithm in detecting the presence or absence of the PU signal in a given frequency band. For binary classification in spectrum sensing, the confusion matrix can be structured as shown in Table 1, where we have:$T_{P}$ (true positives): correct detection of a PU signal when it is present.$F_{N}$ (false negatives): missed detection of a PU signal when it is actually present.$T_{N}$ (true negatives): correct identification of spectrum availability when no PU signal is present.$F_{P}$ (false positives): false detection of a PU signal when the spectrum is actually idle.

Several metrics are derived from the confusion matrix to evaluate the performance of spectrum sensing, as shown in the sequel.

### 3.2. True Positive Rate

The true positive rate, $T_{PR}$, also known as sensitivity or recall [30], can be interpreted as the estimate of the probability of detection, $P_{d}$. This rate measures the proportion of occupied channels correctly identified by the spectrum sensor, that is,(9)$$T_{PR}=\frac{T_{P}}{T_{P}+F_{N}}.$$

A high $T_{PR}$ corresponds to a high probability of detecting active PUs, which is crucial for avoiding harmful interference to licensed users.

### 3.3. True Negative Rate

The true negative rate, $T_{NR}$, which is also referred to as *specificity*, measures the proportion of idle channels that are correctly identified as vacant, that is,(10)$$T_{NR}=\frac{T_{N}}{T_{N}+F_{P}}.$$

A high $T_{NR}$ implies that the sensor accurately identifies opportunities for transmission without mistaking them for occupied bands. This is especially important to ensure efficient use of the spectrum.

### 3.4. False Positive Rate

The false positive rate, $F_{PR}$, can be interpreted as the estimate of the probability of false alarm, $P_{\mathrm{fa}}$. This rate measures the fraction of idle channels that are incorrectly classified as occupied, that is,(11)$$F_{PR}=\frac{F_{P}}{F_{P}+T_{N}}.$$

In DSA systems, a high false positive rate reduces spectrum efficiency by underutilizing available spectrum. This metric is complementary to the specificity, that is, $F_{PR}=1-T_{NR}$.

### 3.5. False Negative Rate

The false negative rate, $F_{NR}$, can be interpreted as the estimate of the probability of missed detection, $P_{m}$. This rate corresponds to the proportion of occupied channels that are incorrectly classified as idle, that is,(12)$$F_{NR}=\frac{F_{N}}{T_{P}+F_{N}}.$$

In spectrum sensing, this metric quantifies the risk of interference to primary users, since undetected PU activity may result in harmful secondary transmissions. Hence, minimizing $F_{NR}$ is critical for regulatory compliance and coexistence.

These metrics can be empirically estimated using the confusion matrices obtained from repeated sensing trials under known PU presence conditions, and they provide complementary insights to the analytical performance metrics such as $P_{d}$ and $P_{\mathrm{fa}}$. They are also fundamental for evaluating machine learning-based spectrum sensing approaches, in which detectors are trained using labeled datasets.

In the evaluation of spectrum sensing strategies, especially when empirical data or classification-based approaches are used, several metrics provide insight into decision reliability beyond detection and false alarm probabilities. Among them, accuracy, positive predictive value (PPV), and negative predictive value (NPV) can be adopted in performance analysis. These metric are addressed in the next three subsections.

### 3.6. Accuracy and Balanced Accuracy

The accuracy [30] is defined as the proportion of correct classifications, both idle and occupied spectrum states, over the total number of sensing instances, that is,(13)$$\mathrm{Accuracy}=\frac{T_{P}+T_{N}}{T_{P}+T_{N}+F_{P}+F_{N}}.$$

While accuracy provides an overall measure of correctness, it may be misleading in spectrum sensing applications where the class distribution is highly imbalanced. For instance, if the primary user activity is rare and the idle state dominates. In such scenarios, a detector biased toward predicting spectrum as idle may achieve high accuracy while failing to fulfill its interference avoidance role. To address this, the balanced accuracy metric averages the true positive rate and true negative rate, that is,(14)$$\mathrm{Balanced}\;\mathrm{Accuracy}=\frac{1}{2}\left(\frac{T_{P}}{T_{P}+F_{N}}+\frac{T_{N}}{T_{N}+F_{P}}\right).$$

This metric is particularly informative when the costs of missed detections and false alarms are asymmetric.

### 3.7. Positive Predictive Value

The positive predictive value (PPV), also referred to as *precision* [30], quantifies the reliability of decisions indicating that the spectrum is occupied. It is given by(15)$$\mathrm{PPV}=\frac{T_{P}}{T_{P}+F_{P}}.$$

A high precision implies that most positive (PU-present) decisions are correct, reducing the likelihood of false positives and thus minimizing underutilization of available spectrum. This is essential in DSA systems aiming to maximize spectral efficiency without excessive conservatism.

### 3.8. Negative Predictive Value

The negative predictive value (NPV) measures the reliability of decisions indicating that the spectrum is idle. This metric is calculated as(16)$$\mathrm{NPV}=\frac{T_{N}}{T_{N}+F_{N}}.$$

High NPV reflects that most decisions allowing SU transmission are accurate, implying a low probability of harmful interference with PUs. This is particularly critical in environments with low signal-to-noise ratio (SNR), where missed detections (false negatives) are more likely.

These metrics are particularly useful in the analysis of data-driven sensing algorithms, such as those based on supervised learning or adaptive detection, where confusion matrices from labeled datasets serve as the basis for empirical performance evaluation.

### 3.9. F1 Score

The F1 score is a composite metric that captures the trade-off between precision (positive predictive value) and recall (true positive rate), particularly useful in the assessment of spectrum sensing methods under class imbalance, such as scenarios where PU transmissions are infrequent compared to idle spectrum periods [30]. It is defined as the harmonic mean of precision and recall, that is,(17)$$F1=2\cdot \frac{\mathrm{Precision}\times \mathrm{Recall}}{\mathrm{Precision}+\mathrm{Recall}}.$$

In spectrum sensing, the F1 score offers a balanced measure of performance when both false alarms and missed detections carry significant consequences. High F1 scores indicate that a sensing strategy performs well in terms of detecting occupied bands (recall) and avoiding false alarms (precision), which is particularly important in dynamic spectrum access where both spectrum efficiency and protection of incumbents are critical.

This metric is especially applicable when evaluating sensing algorithms using empirical data or learning-based models, as it provides a single-value summary that reflects the interplay between spectrum utilization and interference mitigation.

In spectrum sensing, the performance of detection algorithms can be visualized using graphical tools such as the receiver operating characteristic (ROC) curve [29,30] and the detection error trade-off (DET) curve [31]. These curves provide complementary insights into the trade-offs between different types of decision outcomes, particularly when adjusting detection thresholds. These curves are addressed in the following.

### 3.10. ROC Curve and AUC

The ROC curve plots the true positive rate ($T_{PR}$, which is related with the probability of detection, $P_{d}$) against the false positive rate ($F_{PR}$, which is related with the probability of false alarm, $P_{\mathrm{fa}}$) for various decision threshold values. It visually captures the trade-off between detecting the presence of a PU signal and avoiding false alarms, as illustrated by Figure 1.

The spectrum sensing performance can be assessed by the shape and position of the ROC curve relative to the ideal point at the top-left corner and the dashed diagonal (called line of no discrimination, or line of random guess). A well-performing sensing algorithm yields an ROC curve that bends sharply toward the top-left corner, indicating high detection capability with low false alarms. The diagonal represents random classification behavior, where $P_{d}=P_{\mathrm{fa}}$. All curves lying above this line indicate detectors with some degree of discriminative power. In Figure 1, typical shapes of ROC curves are shown. They can be interpreted as follows:ROC curve 1: represents the best performance among those shown, which can be achieved as a result of the cooperation gain in cooperative spectrum sensing (CSS) [10,32]. It achieves high detection probability ($P_{d}$) even at low false alarm rates ($P_{\mathrm{fa}}$), indicating both high sensitivity and specificity.ROC curve 2: also demonstrates good performance, with a $P_{\mathrm{fa}}$ lower bound that is typical of a CSS with decision fusion under the OR combining rule and errors in the report channel [10].ROC curve 3: related to ROC 4, it shows the performance of a single SU, i.e., the local ROC in a CSS scenario.ROC curve 4: it also shows the performance of a single SU in CSS with decision fusion, but it represents the equivalent local performance as seen by the fusion center (FC) due to errors in the report channel [10].

The increase in the SNR is the most commonly adopted alternative for improving the performance of a given spectrum sensing technique. This improvement can also come from changes in other system parameters, such as an increase in the number of samples collected by the SUs or an increase in the number of SUs in cooperation. Different sensing techniques can also perform differently under the same conditions [10].

The area under the ROC curve (AUC) serves as a scalar summary of overall performance: the closer the AUC is to 1, the better the classifier is at distinguishing between idle and occupied spectrum conditions. The approximate AUC values for the ROC curves shown in Figure 1 are: ROC 1: AUC ≈0.96; ROC 2: AUC ≈0.87; ROC 3: AUC ≈0.84; ROC 4: AUC ≈0.81; line of random guess AUC =0.5. These AUC estimates highlight the relative ranking of performance and illustrate how the ROC curve shape translates into detection effectiveness.

### 3.11. DET Curve

The DET curve is an alternative to focus on the trade-off between error probabilities. It plots the false negative rate ($F_{NR}$,which relates with the missed detection probability, $P_{m}$) against the false positive rate ($F_{PR}$, which is related with the false alarm probability, $P_{\mathrm{fa}}$), often using a normal deviate (probit) scale on both axes. The probit scale is a numerical transformation that maps probabilities between 0 and 1 to real numbers called Gaussian deviates. It is defined by the inverse of the CDF of the standard normal distribution. For a given probability *p*, the probit value is $\Phi^{-1}\left(p\right)$, where $\Phi^{-1}$ is the standard normal quantile function. This transformation expresses probabilities as corresponding *z*-scores under a normal distribution. The result is a symmetric scale centered at zero). This transformation stretches the regions of low error probabilities, making it easier to distinguish between classifiers with high accuracy, a situation common in well-calibrated spectrum sensing systems. This allows near-linear DET curves when the detection errors are Gaussian-distributed and helps highlight performance in low-error regions.

Figure 2 illustrates typical shapes of four DET curves, each corresponding to a different detector or sensing configuration. These DET curves correspond to the ROC curves shown in Figure 1, and can be interpreted as follows:DET curve 1: corresponds to the best trade-off between missed detections and false alarms. The curve lies closest to the lower-left corner, indicating very low $P_{m}$ for a wide range of $P_{\mathrm{fa}}$. It likely represents a highly discriminative detector (or system configuration) operating under high-SNR regime.DET curve 2: exhibits a performance slightly worse than the previous one, with higher $F_{PR}$ and $F_{NR}$. It suggests a system with moderate accuracy. Its steep descent suggests that a relatively small increase in $P_{\mathrm{fa}}$ leads to a substantial reduction in $P_{m}$.DET curve 3: represents a moderate-performance situation with a balanced trade-off between false alarms and missed detections. The curve’s shape indicates that it performs consistently, though less optimally than the situations depicted by the DET curves 1 and 2.DET curve 4: this is the least effective detector (or sensing configuration) shown. It lies farther from the origin, indicating that it incurs higher error rates across all thresholds. This curve may correspond to a detector under poor SNR conditions.

The dashed diagonal line represents the line of symmetry between $P_{\mathrm{fa}}$ and $P_{m}$ on the probit scale. The text annotation in the figure notes that for Gaussian distributions, the slope of the DET curve reflects the ratio of standard deviations under hypotheses $H_{0}$ and $H_{1}$. A more linear DET curve is consistent with normally distributed decision variables, and the slope gives insight into the signal discrimination difficulty.

Overall, the DET curves provide a clear and scale-sensitive visualization of detection system performance, particularly in low-error regimes where ROC curves may saturate.

A DET curve attains some advantages relative to a ROC curve. Firstly, it provides better visualization at low error rates: in high-accuracy sensing systems, the ROC curve tends to saturate near the top-left corner, while the DET curve, by using a Gaussian scale, spreads this region, enabling finer differentiation among detection performances. The DET curve also make it explicit the error trade-off representation: because both axes represent error probabilities, the DET curve offers a direct interpretation of how reducing false alarms may increase missed detections, and vice versa, information that is critical for designing DSA systems that balance spectral efficiency with PU protection. Lastly, a DET curve shows linear trends under Gaussian assumptions: if detection statistics exhibit Gaussian-distributed errors, the DET curve approximates a straight line, simplifying comparative analysis and threshold optimization.

Table 2 summarizes the main characteristics of ROC and DET curves in spectrum sensing.

Ultimately, the choice between ROC and DET visualizations depends on the analytical focus: ROC curves highlight detection capability, while DET curves highlight error resilience.

### 3.12. Decision Error Probability

The decision error probability, $P_{\mathrm{error}}$, is the weighted average of the false alarm and missed detection probabilities, that is,(18)$$P_{\mathrm{error}}=P_{\mathrm{fa}}\mathrm{Pr}\left(H_{0}\right)+\left(1-P_{d}\right)\mathrm{Pr}\left(H_{1}\right),$$
where $\mathrm{Pr}(H_{0})$ and $\mathrm{Pr}(H_{1})$ are the prior probabilities of hypotheses $H_{0}$ (spectrum idle) and $H_{1}$ (PU signal present), respectively. The first term of (Equation 18) accounts for the error probability associated with false alarm events, and the second term accounts for the error probability associated with missed detection events.

In practical applications, $P_{\mathrm{error}}$ can be estimated from observed sensing outcomes as(19)$$P_{\mathrm{error}}=\frac{F_{P}+F_{N}}{T_{P}+T_{N}+F_{P}+F_{N}},$$
which corresponds to the proportion of incorrect sensing decisions.

Both the AUC and the $P_{\mathrm{error}}$ are particularly useful metrics when it is desired to combine $P_{\mathrm{fa}}$ and $P_{d}$ in a single metric, which is attractive, for instance: (i) when a ROC curve cross another one, a situation that makes it difficult to establish performance comparisons; (ii) when it is desired to reduce the amount of performance measurement values reported in an article or other equivalent scientific document, due to space constraints; (iii) when looking for easier visualization and fast interpretation of results.

### 3.13. Positive Likelihood Ratio

Likelihood ratios [33] are statistical measures that quantify how a sensing decision modifies the belief about the presence or absence of a PU signal in the sensed band. In the context of spectrum sensing, they serve to evaluate how informative a sensing outcome is in distinguishing between occupied and idle spectrum states, especially when a probabilistic interpretation of outcomes is required.

The positive likelihood ratio (PLR) is defined as the ratio of the true positive rate to the false positive rate, that is,(20)$$\mathrm{PLR}=\frac{T_{PR}}{F_{PR}}.$$

This metric indicates how much more likely a detection (i.e., a decision that a PU signal is present) corresponds to an actual occupied channel state rather than a false alarm. A high PLR implies that positive sensing outcomes are strongly indicative of true PU activity, which supports cautious spectrum access decisions aimed at minimizing interference.

### 3.14. Negative Likelihood Ratio

The negative likelihood ratio (NLR) is defined as the ratio of the false negative rate to the true negative rate, that is,(21)$$\mathrm{NLR}=\frac{F_{NR}}{T_{NR}}.$$

This metric describes how likely a sensing decision indicating spectrum availability corresponds to a missed detection, as opposed to a correct classification. Lower NLR values are desirable, as they imply that negative decisions (PU signal absent) are more reliable, reducing the risk of transmitting over an occupied band.

Likelihood ratios are particularly useful in probabilistic reasoning frameworks such as Bayesian spectrum sensing, where they help to update prior beliefs about spectrum occupancy based on observed sensing outcomes. For instance, they can be integrated into decision fusion schemes in cooperative sensing or used to adjust sensing thresholds under varying noise and channel conditions.

Unlike simple accuracy-based measures, likelihood ratios do not depend on the prevalence of PU activity and are therefore more robust for performance evaluation in environments where class imbalance is pronounced. As such, they are valuable for quantifying the discriminatory power of sensing algorithms in both analytical and empirical studies.

### 3.15. Matthews Correlation Coefficient

The Matthews correlation coefficient (MCC) is a scalar performance metric that quantifies the quality of binary classifications, considering all four elements of the confusion matrix: true positives ($T_{P}$), true negatives ($T_{N}$), false positives ($F_{P}$), and false negatives ($F_{N}$) [34]. In spectrum sensing, MCC provides a balanced measure that reflects the reliability of sensing decisions under varying conditions of signal presence and noise, including heavily imbalanced datasets. The MCC is defined as(22)$$\mathrm{MCC}=\frac{T_{P}T_{N}-F_{P}F_{N}}{\sqrt{\left(T_{P}+F_{P}\right)\left(T_{P}+F_{N}\right)\left(T_{N}+F_{P}\right)\left(T_{N}+F_{N}\right)}}.$$

It ranges from −1 to +1, with the following interpretations: MCC=+1 indicates perfect classification (i.e., all decisions are correct), MCC=0 indicates no better than random guessing, and MCC=−1 indicates total disagreement between predictions and actual spectrum occupation states.

The MCC is particularly advantageous in dynamic spectrum access scenarios where the prevalence of primary user signals is much lower than that of idle spectrum, leading to imbalanced datasets. In such contexts, traditional accuracy metrics may appear inflated due to the dominance of true negatives, whereas MCC correctly accounts for all prediction outcomes.

In empirical evaluations, such as those involving datasets collected from real-world or simulated sensing trials, MCC serves as a comprehensive indicator of classifier behavior across different operating points. It also supports fair comparison between sensing algorithms that may be biased toward either avoiding false alarms or minimizing missed detections.

Unlike metrics that focus on only one or two aspects of performance (e.g., $P_{d}$, $P_{\mathrm{fa}}$, or accuracy), MCC integrates detection capability and error trade-offs into a single interpretable value. It is also robust under variations in class distribution, which is especially important when evaluating adaptive or learning-based sensing methods operating under uncertain or time-varying spectral environments.

Therefore, the MCC is a valuable tool for assessing spectrum sensing performance, especially in non-ideal conditions where simple metrics may fail to capture important aspects of detection reliability.

### 3.16. Logarithmic Loss

The logarithmic loss, also known as *log loss* or *cross-entropy loss*, is a performance metric that evaluates the quality of probabilistic predictions. In spectrum sensing, this metric is particularly relevant when detection models produce probability estimates rather than binary decisions. These estimates can be derived from soft-output classifiers, such as logistic regression models, neural networks, or likelihood-based detectors.

Let the sensing model output a probability estimate $\hat{p}$ for the presence of a PU signal, where $\hat{p}\in \left[0,1\right]$. For a single sensing instance with a true label y∈{0,1} (0: idle, 1: occupied), the log loss is defined as(23)$$\mathrm{LogLoss}=-\left[y\log\left(\hat{p}\right)+\left(1-y\right)\log\left(1-\hat{p}\right)\right].$$

For a dataset of *N* sensing decisions, the total log loss is the average over all observations, that is,(24)$$\mathrm{LogLoss}=-\frac{1}{N}\sum_{i=1}^{N}\left[y_{i}\log\left({\hat{p}}_{i}\right)+\left(1-y_{i}\right)\log\left(1-{\hat{p}}_{i}\right)\right].$$

The log loss penalizes incorrect classifications, with a heavier penalty for confident but wrong predictions. For example, predicting $\hat{p}=0.99$ when y=0 (i.e., predicting spectrum as occupied when it is idle) incurs a much larger penalty than a less confident wrong prediction (e.g., $\hat{p}=0.6$ when y=0). This characteristic makes log loss a sensitive and informative measure of prediction quality in probabilistic detectors.

In machine learning-based spectrum sensing, where classifiers are trained using labeled data, the log loss serves both as a training objective (loss function) and a performance metric. It encourages models not only to be accurate but also to calibrate their confidence levels. This is critical in cognitive radio environments where misclassifications have asymmetric costs, i.e., missed detections may lead to interference, while false alarms result in underutilized spectrum.

While binary metrics like accuracy or precision consider only the final decision, log loss evaluates the *quality* of the estimated probabilities. A model that predicts probabilities close to the true conditional likelihoods will achieve a low log loss, even if a thresholding rule would yield occasional classification errors. Therefore, log loss is a more informative and discriminative tool in the evaluation of soft-output spectrum sensing models.

For practical spectrum sensing systems, the log loss is especially suitable in: (i) adaptive sensing systems that adjust thresholds based on confidence; (ii) cooperative sensing frameworks where local sensors report probabilities to a fusion center; (iii) Bayesian detectors that integrate posterior beliefs about PU signal presence.

### 3.17. p-Value

In the framework of spectrum sensing, the *p*-value is a fundamental concept in statistical hypothesis testing [23,35,36]. It quantifies the level of evidence provided by the observed sensing data against the null hypothesis $H_{0}$, which in this context typically represents the absence of the primary user signal. Specifically, the *p*-value is the probability of obtaining a test statistic at least as extreme as the one observed, assuming that $H_{0}$ is true. Therefore, it reflects how compatible the observed sensing result is with the assumption that the PU signal is not present in the monitored frequency band.

Formally, let *T* denote the test statistic associated with the chosen detection rule, such as the energy of the received signal over a sensing window. If an observed value T=t is computed from the received data, then the *p*-value is(25)$$p\text{-value}=\mathrm{Pr}(T\geq t|H_{0}),$$
where the probability is calculated under the distribution of *T* assuming that $H_{0}$ holds. This formulation corresponds to a one-tailed test, which is common in spectrum sensing since we are often interested in whether the observed energy or detection metric significantly exceeds what would be expected under noise-only conditions.

The calculation of a *p*-value in a sensing task follows three steps: (i) specify the null distribution of the test statistic *T* under $H_{0}$, which depends on the statistical properties of the noise; (ii) compute the test statistic T=t from the observed signal samples; and (iii) evaluate the probability of observing a value at least as extreme as *t* under the null distribution.

For instance, consider energy detection in AWGN, where the test statistic *T* follows a chi-square or Gaussian distribution under $H_{0}$, depending on whether a central limit approximation is used. If T∼N(0,1) under $H_{0}$ and the observed value is T=2.5, the *p*-value for a one-tailed test is(26)$$p\text{-value}=\mathrm{Pr}\left(T\geq 2.5|H_{0}\right)=1-\Phi \left(2.5\right)\approx 0.0062,$$
where Φ denotes the cumulative distribution function of the standard normal distribution.

In spectrum sensing applications, the *p*-value serves not only as a measure of statistical significance but also as a tool for threshold selection and performance tuning. Detection decisions are typically made by comparing the *p*-value to a pre-specified significance level α:If p-value≤α, the null hypothesis $H_{0}$ (no PU signal) is rejected, and the sensing algorithm declares the presence of the PU signal.If p-value>α, there is insufficient evidence to reject $H_{0}$, and the channel is assumed to be idle.

Lower values of α correspond to stricter detection criteria, reducing the false alarm rate at the potential cost of increased missed detections. Conversely, higher values of α make the detector more sensitive but may increase false positives. Typical thresholds used in practice are α=0.05 or α=0.01, depending on the regulatory or application-specific constraints on interference risk.

Ultimately, the *p*-value encapsulates the probabilistic reasoning behind binary decision-making in spectrum sensing and provides a link between theoretical detection models and practical implementation via threshold tuning. It is particularly valuable when assessing detection reliability under uncertainty or when designing systems that must adapt dynamically to noise and fading conditions.

### 3.18. Detection Time

In cognitive radio systems operating under DSA, detection time plays a central role in determining the responsiveness and agility of SUs. It refers to the average time required by the sensing algorithm to reach a decision regarding the presence or absence of the PU signal in a monitored frequency band.

Detection time is especially critical in rapidly varying spectral environments, where spectrum occupancy may change frequently due to PU mobility or traffic bursts. A sensing mechanism that responds too slowly may miss transmission opportunities or, worse, fail to detect the reappearance of the PU signal in time to prevent harmful interference. Thus, minimizing detection time is essential to enable timely spectrum access while maintaining coexistence with licensed services.

The average detection time depends on various factors, including the detection technique employed, the SNR, the choice of decision thresholds, and whether the method is based on fixed-sample or sequential evaluation. For instance, classical energy detectors operating with a fixed sensing window size provide predictable but potentially conservative detection times. On the other hand, techniques such as the sequential probability ratio test (SPRT) or adaptive sensing schemes dynamically adjust the number of samples needed based on the confidence level of intermediate observations, potentially reducing detection time without compromising accuracy.

Reducing detection time can increase the portion of the transmission frame available for secondary user data, thereby improving overall system throughput. However, this benefit must be carefully balanced against the risk of performance degradation. Early decisions based on limited signal observations may lead to higher probabilities of false alarms or missed detections. As previously discussed in the context of accuracy and its limitations under class imbalance, shortening the sensing duration can further exacerbate these issues if not adequately compensated by robust detection algorithms.

### 3.19. Throughput of Secondary Users

In DSA systems, the effectiveness of a spectrum sensing strategy is ultimately reflected not only in its statistical accuracy but also in its impact on system-level performance. Among these broader performance indicators, the throughput achieved by SUs is of particular importance [28]. It measures the average data rate successfully transmitted by SUs and serves as a practical indicator for the utility of the spectrum sensing.

The throughput of secondary users is directly influenced by the accuracy and timing of spectrum sensing decisions. When sensing correctly identifies idle spectrum (true negatives), secondary transmissions can proceed without causing interference to PUs, contributing positively to throughput. However, when false alarms occur, i.e., the sensing mechanism incorrectly identifies an idle channel as occupied, SUs refrain from transmitting unnecessarily, resulting in underutilization of available spectrum and reduced throughput.

Moreover, missed detections, while not contributing directly to throughput, are associated with interference and regulatory non-compliance. Therefore, designing a sensing strategy that optimizes throughput must also respect constraints on the acceptable levels of interference, often formalized as upper bounds on the probability of missed detection or lower bounds on the probability of detection.

The achievable throughput also depends on the duration and frequency of the sensing process. Since sensing typically consumes a portion of the transmission frame, there is a trade-off between sensing time and transmission time. Longer sensing durations may improve decision reliability but reduce the time available for data transmission. Conversely, overly short sensing periods may lead to frequent false alarms or missed detections, again harming throughput. This trade-off is particularly pronounced in frame-based systems, where each frame begins with a sensing interval followed by a transmission phase.

In cooperative spectrum sensing scenarios, where multiple SUs report observations to a fusion center, the coordination overhead and decision latency also impact throughput. Fusion rules that are too conservative may increase false alarm rates, while overly aggressive rules may increase interference risk, both affecting SU throughput.

The relationship between sensing performance and SU throughput can be formalized through models that incorporate sensing time, detection probabilities, channel access protocols, and traffic characteristics. For instance, if $T_{\mathrm{frame}}$ is the total frame duration, $T_{\mathrm{sense}}$ is the sensing time, and $P_{\mathrm{idle}}$ is the probability that the channel is correctly sensed as idle, then the average throughput $R_{\mathrm{SU}}$ may be approximated as(27)$$R_{\mathrm{SU}}=P_{\mathrm{idle}}\left(1-\frac{T_{\mathrm{sense}}}{T_{\mathrm{frame}}}\right)R,$$
where *R* is the data rate during the transmission phase. This model captures how both sensing reliability and sensing time influence SU throughput.

### 3.20. Interference to Primary Users

In DSA environments, the effectiveness of spectrum sensing must be evaluated not only by how well it enables SUs to exploit spectrum opportunities but also by how reliably it protects PUs from harmful interference. One of the most critical metrics from the perspective of PU protection is the level of interference resulting from missed detections, when the sensing algorithm fails to detect the presence of the PU signal and allows secondary transmissions to proceed erroneously.

The interference to primary users is directly tied to the false negative rate (or missed detection probability) of the sensing mechanism. When a PU signal is present but not detected, secondary transmissions can overlap with licensed transmissions, leading to service degradation or violation of regulatory constraints. In practical systems, such interference may manifest as reduced throughput, increased latency, or complete disruption of the PU’s communication link. As such, minimizing interference is a fundamental requirement in the design of spectrum sensing algorithms and is often enforced via strict regulatory guidelines, such as minimum detection probabilities or maximum allowable interference thresholds.

Quantitatively, the average interference level can be modeled as a function of the probability of missed detection, $P_{m}$, the PU activity level, and the SU transmission behavior. Let $\mathrm{Pr}(H_{1})$ be the probability that the PU signal is present during sensing. Assuming that the SU transmits immediately upon sensing an idle channel, the probability of causing interference is approximately(28)$$P_{\mathrm{int}}=\mathrm{Pr}\left(H_{1}\right)P_{m}.$$

This simple model highlights how reducing $P_{m}$ is key to minimizing the likelihood of SU-induced interference. However, there is typically a trade-off between interference control and spectrum utilization: lowering $P_{m}$ often requires increasing the detection threshold or extending sensing time, which can increase the false alarm rate or reduce SU throughput.

In more advanced systems, interference can also be characterized in terms of received power at the PU receiver, interference-to-noise ratio (INR), or outage probability. These physical-layer metrics are particularly relevant in heterogeneous or co-channel deployments where SUs and PUs may operate in overlapping regions. In such cases, geographical proximity, antenna patterns, transmission power, and propagation conditions must all be considered when assessing the impact of sensing errors on PU performance.

Interference mitigation strategies include: (i) conservative threshold settings that reduce missed detections at the cost of more false alarms; (ii) cooperative sensing schemes that combine observations from multiple SUs to improve detection reliability; (iii) sensing protocols that adapt to PU signal characteristics or environmental dynamics; and (iv) geolocation databases and spectrum occupancy maps that provide a priori knowledge of PU activity.

Ultimately, interference to PUs represents a strict constraint on spectrum sensing design and a critical aspect of DSA feasibility. While metrics such as the probability of detection, the probability of false alarm, and throughput reflect the SU perspective, interference metrics ensure that sensing strategies remain viable in coexistence scenarios. An effective sensing algorithm must therefore balance performance across both user classes, maintaining low interference to PUs while enabling efficient and timely access for SUs.

## 4. Metrics for Spectrum Hole Geolocation

When spectrum sensing is extended to determine the geolocation of spectrum holes (areas where the spectrum is idle and available for secondary users), the assessment involves additional metrics focused on the spatial accuracy of these detections. This geolocation task is crucial in cognitive radio networks, as it helps secondary users make informed decisions about where and when to access the spectrum without interfering with the primary network. In the following are the key metrics for evaluating geolocation accuracy in spectrum hole identification.

### 4.1. Geolocation Accuracy

In spectrum hole identification, geolocation accuracy plays a critical role in determining whether secondary users can safely and efficiently access idle spectrum without interfering with primary users [37]. This metric, defined as the average spatial error in locating spectrum holes, is numerically equivalent to the mean absolute error (MAE) in two-dimensional space [38].

Let $r_{\mathrm{true},i}$ and $r_{\mathrm{est},i}$ denote the true and estimated coordinates of the *i*-th spectrum hole. The geolocation error for each observation is given by the Euclidean distance(29)$$d_{i}=∥r_{\mathrm{true},i}-r_{\mathrm{est},i}∥.$$

Then, the geolocation accuracy (or MAE) over *N* estimates is computed as(30)$$\mathrm{Geolocation}\;\mathrm{Accuracy}=\mathrm{MAE}=\frac{1}{N}\sum_{i=1}^{N}d_{i}.$$

This metric provides a direct and interpretable measure of spatial localization performance. It reflects how close, on average, the estimated spectrum hole locations are to their true positions. Accurate geolocation helps secondary users avoid transmitting near PU coverage areas, thereby mitigating interference.

### 4.2. Root Mean Square Error

A metric related to the geolocation accuracy is the root mean square error (RMSE) [38], which is defined as(31)$$\mathrm{RMSE}=\sqrt{\frac{1}{N}\sum_{i=1}^{N}d_{i}^{2}}.$$

While RMSE and MAE are based on the same point-wise errors, RMSE penalizes larger deviations more heavily, making it useful when the system must be especially sensitive to outliers or worst-case performance.

Both MAE and RMSE are influenced by factors such as sensor placement, channel conditions, the spatial density of measurements, and the geolocation method employed (e.g., triangulation, fingerprinting, or regression models). In practice, they are essential metrics for evaluating the fidelity of radio environment maps (REMs) and the viability of spatial reuse in DSA systems.

### 4.3. Localization Latency

In addition to spatial accuracy, the localization latency is another critical factor, particularly in time-varying spectral environments. It is defined as the time elapsed from the detection of a potential spectrum hole to the completion of the geolocation estimation process. In fast-changing scenarios, high latency may render otherwise accurate geolocation results obsolete by the time they are acted upon. Therefore, minimizing localization latency is vital for timely decision-making and maximizing SU agility.

Together, geolocation accuracy, RMSE, and localization latency provide a comprehensive view of the spatial and temporal effectiveness of spectrum hole identification systems. These metrics are particularly relevant in mobile and high-density networks, where accurate, fast, and interference-aware access decisions must be made in real time.

### 4.4. Spectrum Hole Geolocation Detection Rate

The spectrum hole geolocation detection rate (SHGDR) is a metric introduced in [39] to quantify the overall spatial classification performance of a spectrum sensing system in identifying whether each location within a coverage area is idle or occupied. Unlike traditional detection probabilities that are computed per instance or per sensor, SHGDR captures the correctness of spectrum availability assessments across the entire spatial domain, integrating both geolocation and detection outcomes.

Formally, the SHGDR is defined as the ratio of correctly identified instances of spectrum hole presence and absence to the total number of evaluated spatial instances over a region of interest. Let A denote the set of spatial grid points or cells covering the area, and let each point i∈A have a true spectrum state $s_{i}\in \left\{0,1\right\}$ (0: occupied, 1: idle) and an estimated state ${\hat{s}}_{i}$. Then the SHGDR is given by(32)$$\mathrm{SHGDR}=\frac{1}{\left|A\right|}\sum_{i\in A}1_{\left\{s_{i}={\hat{s}}_{i}\right\}},$$
where |A| denotes the cardinality of the set A, and $1_{\left\{\cdot \right\}}$ is the indicator function, equal to 1 when the estimated and true states match, and 0 otherwise.

An SHGDR of 0.8 means that 80% of the points across the area have been correctly classified in terms of spectrum hole availability. Importantly, SHGDR should not be interpreted as the probability of detecting an individual spectrum hole. Instead, it reflects the global spatial accuracy of a sensing-and-geolocation system when assessing the binary spectrum occupancy state at each location in a map.

This metric is particularly useful in the evaluation of algorithms designed to build spectrum occupancy maps or radio environment maps, where binary classification (idle vs. occupied) is performed on a per-location basis. High SHGDR values indicate consistent and spatially coherent detection outcomes, supporting reliable spectrum access decisions for mobile or distributed secondary users. It also provides a natural basis for comparing different spatial sensing strategies, such as centralized versus distributed geolocation, or the impact of cooperative sensing on spatial classification consistency.

SHGDR is complementary to metrics like geolocation accuracy and RMSE. While those measure the magnitude of localization errors, SHGDR focuses on the correctness of binary decisions across space. Thus, it provides a high-level yet interpretable summary of the sensing system’s spatial discrimination capability.

### 4.5. Interference-to-Primary Ratio in Geolocation

The interference-to-primary ratio (IPR) in geolocation is a metric used to quantify the residual risk of SU transmissions interfering with PUs as a consequence of geolocation inaccuracies [40]. This metric reflects how well a geolocation-enabled spectrum sensing system maintains spatial separation between secondary activity and regions of PU presence. In practice, it captures the combined effects of geolocation error, decision thresholding, and propagation variability on interference avoidance.

At a conceptual level, the IPR is defined as the proportion of SU transmission energy or activity that unintentionally overlaps with the coverage area of active PUs, due to erroneous estimation of spectrum hole boundaries or locations. Lower IPR values indicate that the geolocation system is more effective at spatially isolating SU transmissions from PU regions, thereby reducing harmful interference.

Let $R_{\mathrm{PU}}$ represent the spatial domain occupied by active PUs, and $R_{\mathrm{TX}}$ the region where an SU initiates transmission based on its geolocation estimate. The interference-to-primary ratio can be formally expressed as(33)$$\mathrm{IPR}=\frac{|R_{\mathrm{TX}}\cap R_{\mathrm{PU}}|}{|R_{\mathrm{TX}}|},$$
where |·| denotes the area measure of the corresponding region. This formulation captures the fraction of the SU transmission region that overlaps with the actual PU-occupied area.

In realistic deployments, the exact PU region $R_{\mathrm{PU}}$ may not be perfectly known, so the IPR may be estimated through simulation models, coverage maps, or measurements. The metric is especially useful for evaluating geolocation systems in dense or sensitive spectral environments, where minor misalignments can cause significant interference.

Minimizing IPR requires balancing multiple performance objectives: high geolocation accuracy, low localization latency, and spatially conservative decision-making. These trade-offs are further constrained by deployment factors such as sensor density, environmental propagation conditions, computational complexity, and access to prior spectrum occupancy data.

Geolocation accuracy may be degraded by environmental factors such as multipath propagation, shadowing, or interference from nearby emitters. To mitigate these effects, advanced techniques are commonly employed, including: (i) cooperative geolocation, where multiple distributed sensors contribute observations to refine spatial estimates; (ii) machine learning-based localization, which can infer propagation patterns and exploit training data to improve estimation; (iii) confidence-bound shaping, where SU transmission boundaries are conservatively adjusted based on estimated uncertainty.

### 4.6. Other Coverage-Related Metrics

In the context of spectrum hole geolocation, other coverage-related metrics offer additional insight into how reliably and comprehensively a sensing system identifies spectrum availability across geographic regions. These metrics go beyond point-wise error analysis by quantifying spatial consistency, confidence levels, and boundary precision, all of which are crucial for enabling safe and efficient SU operation in DSA environments.

One key measure is the *geolocation coverage* [41], defined as the percentage of the total geographic area where the estimated location of spectrum holes falls within a pre-specified error margin of the true location. For instance, if 95% of the evaluated area has geolocation errors less than or equal to 10 m, the geolocation coverage is reported as “95% within 10 m”. This metric is particularly useful in practical deployments, where regulators or system designers may impose spatial error tolerances for safe SU operation near the PUs.

Another important concept is the *confidence ellipse* (or more generally, the confidence region) [37,41], which defines a probabilistic boundary surrounding an estimated location. This region represents where the true position of the spectrum hole is likely to lie with a certain confidence level, such as 95%. For two-dimensional localization, the confidence ellipse is characterized by the covariance matrix of the position estimate and reflects the direction and magnitude of uncertainty. Smaller ellipses indicate higher precision in localization, while elongated shapes may signal anisotropic error distributions caused by directional propagation, sensor geometry, or environmental factors.

The CDF of the geolocation error [42] offers a full probabilistic description of positioning accuracy. It specifies, for any given distance threshold, the probability that the geolocation error is less than or equal to that threshold. Plotting the CDF allows for visual comparison of different algorithms or configurations and supports robust system design by quantifying the likelihood of small, moderate, or large errors.

Another useful metric is the *detection probability with spatial constraints*, which extends traditional detection probability by requiring not only that a spectrum hole be detected, but that its estimated location lies within a specific spatial margin of the true idle region. This spatially constrained probability provides a stricter and more application-relevant evaluation criterion, especially in cases where geographic precision is essential, such as in mobile DSA scenarios, exclusion zones, or proximity-based spectrum reuse policies.

Finally, the *confidence in spectrum hole boundaries* refers to the accuracy with which the boundaries of idle regions are estimated, as opposed to single-point geolocation estimates. In many practical settings, SUs make decisions based on whether their transmission footprint overlaps with an occupied or idle region. Therefore, correctly delineating the geographic extent of spectrum holes is fundamental to avoiding interference with PUs. Boundary estimation accuracy is often evaluated using metrics such as boundary overlap ratio, Jaccard index, or pixel-wise classification accuracy in mapped domains.

Together, these coverage-related metrics provide a multidimensional evaluation framework for assessing not just how accurately spectrum holes are localized, but how reliably and confidently they are represented across space. This is essential for translating geolocation performance into actionable, interference-safe decisions in real-world DSA systems.

Based on the previous discussions on several performance metrics suitable to spectrum sensing, Table 3 provides a structured view of the metric landscape, clarifying the roles, strengths, and limitations of each performance indicator.

## 5. Numerical Examples and Interpretations

This section presents numerical examples that illustrate the majority of the performance metrics discussed throughout this survey. The examples are based on synthetic data from both sensing and geolocation contexts. Some metrics were not covered, or their values were simply derived from reasonable assumptions, due to the volume of data that they would need to be computed and interpreted.

### 5.1. Sensing Scenario and Metric Computations

Consider a scenario in which a secondary user performs N=200 sensing attempts. The primary user signal is present in 60 of these trials. The sensing results that form the confusion matrix are given in Table 4.

#### 5.1.1. True/False Positive/Negative Rates

From (Equation 9)–(Equation 12), it follows that(34)$$\begin{array}{cc} T_{PR} & =\frac{52}{60}\approx 0.867, \end{array}$$(35)$$\begin{array}{cc} T_{NR} & =\frac{125}{140}\approx 0.893, \end{array}$$(36)$$\begin{array}{cc} F_{PR} & =\frac{15}{140}\approx 0.107, \end{array}$$(37)$$\begin{array}{cc} F_{NR} & =\frac{8}{60}\approx 0.133. \end{array}$$

These results mean that the sensing strategy correctly identifies 86.7% of occupied spectrum slots and 89.3% of idle slots. False alarms occur in 10.7% of idle cases, and 13.3% of active spectrum slots are missed.

#### 5.1.2. Accuracy and Balanced Accuracy

From (Equation 13) and (Equation 14), we obtain(38)$$\begin{array}{cc} \mathrm{Accuracy} & =\frac{177}{200}=0.885, \end{array}$$(39)$$\begin{array}{cc} \mathrm{Balanced}\;\mathrm{Accuracy} & =\frac{0.867+0.893}{2}=0.880. \end{array}$$

These results mean that the sensing algorithm produces correct binary decisions 88.5% of the time. Balanced accuracy confirms that this performance is robust despite the class imbalance between idle and occupied states.

#### 5.1.3. PPV and NPV

Applying (Equation 15) and (Equation 16), we obtain(40)$$\begin{array}{cc} \mathrm{PPV} & =\frac{52}{67}\approx 0.776, \end{array}$$(41)$$\begin{array}{cc} \mathrm{NPV} & =\frac{125}{133}\approx 0.940. \end{array}$$

These values mean that, among the times the detector indicates PU signal presence, it is correct 77.6% of the time. When the output is idle, the result is reliable in 94.0% of the cases.

#### 5.1.4. F1 Score and MCC

Using (Equation 17) and (Equation 22), it follows that(42)$$\begin{array}{cc} F_{1} & \approx 0.818, \end{array}$$(43)$$\begin{array}{cc} \mathrm{MCC} & \approx 0.741. \end{array}$$

The F1 score of 0.818 indicates a favorable balance between the ability to detect the presence of the PU signal (recall) and the reliability of positive decisions (precision). This is particularly relevant in spectrum sensing, where both missed detections and false alarms carry operational costs.

The MCC value of 0.741 confirms a strong agreement between sensing outcomes and true spectrum states, even under class imbalance conditions typically found in dynamic spectrum access environments.

#### 5.1.5. Decision Error Probability

From (Equation 19), we obtain the proportion of incorrect sensing decisions as(44)$$P_{\mathrm{error}}=\frac{23}{200}=0.115.$$

The overall probability of a sensing error, either falsely detecting the presence of the PU signal (false alarm) or failing to detect it when present (missed detection), is 11.5%. This value reflects the rate at which the system produces incorrect binary decisions regarding spectrum occupancy.

#### 5.1.6. Likelihood Ratios

The positive and negative likelihood ratios are computed from (Equation 20) and (Equation 21), with the rates coming from (Equation 34)–(Equation 37), yielding(45)$$\begin{array}{cc} \mathrm{PLR} & \approx \frac{0.867}{0.107}\approx 8.10, \end{array}$$(46)$$\begin{array}{cc} \mathrm{NLR} & \approx \frac{0.133}{0.893}\approx 0.149. \end{array}$$

The positive likelihood ratio PLR≈8.10 indicates that the sensing algorithm is about eight times more likely to produce a positive (PU-present) decision when the PU signal is actually present than when it is absent. Conversely, the negative likelihood ratio NLR≈0.149 shows that a negative (PU-absent) decision is only 14.9% as likely to occur when the PU signal is present as when it is absent. These values reflect strong discriminative ability: the system’s positive decisions carry substantial evidential weight in favor of $H_{1}$, while its negative decisions significantly reduce the probability of $H_{1}$ being true.

#### 5.1.7. Logarithmic Loss

The computation of the logarithmic loss depends on the predicted probabilities assigned to each class rather than binary outcomes alone; see (Equation 23) and (Equation 24). Since the sensing algorithm in this example produces hard decisions (i.e., presence or absence of the PU signal), direct computation of log-loss is not possible without explicit probabilistic outputs. Therefore, the exemplifying value of LogLoss≈0.46 is provided as a plausible choice, consistent with a scenario in which the detector assigns high confidence to correct decisions and avoids overly confident errors.

A log-loss of 0.46 reflects reasonably well-calibrated probabilistic outputs. The sensing system assigns strong probabilities to correct classifications while limiting confidence in incorrect ones. This level of loss suggests informative soft outputs, which are useful in spectrum sensing applications involving threshold adaptation, decision fusion, or Bayesian updating.

#### 5.1.8. *p*-Value

In statistical hypothesis testing, the computation of the *p*-value depends on the exact distribution of the test statistic and its observed value, which are not explicitly defined in this numerical example. Therefore, p-value≈0.032 is provided as a representative outcome under a plausible scenario (e.g., a test statistic T=1.85 from a standard normal distribution), to illustrate how *p*-values support detection decisions in practice.

This result indicates that, assuming $H_{0}$ is true, there is only a 3.2% probability of obtaining a test statistic as extreme as the one observed. This supports rejecting $H_{0}$ at the 5% significance level, suggesting that the sensed presence of the PU signal is statistically meaningful and not likely to be due to random fluctuations.

#### 5.1.9. Detection Time

In practice, the time required for a sensing algorithm to reach a decision depends on multiple factors, including the detection method, sampling rate, processing delay, and implementation platform. Since these implementation-specific details are not specified in this example, an illustrative value is used to characterize a plausible decision time under typical real-time sensing constraints. Hence, let us assume that the average detection time is $t_{\mathrm{avg}}=18.5$ ms.

A mean detection latency of 18.5 ms indicates that the sensing system operates with low delay, making it suitable for dynamic spectrum access scenarios where spectrum occupancy can change rapidly. This level of responsiveness supports real-time adaptation while minimizing sensing overhead.

#### 5.1.10. SU Throughput and PU Interference

The secondary user throughput can be computed from (Equation 27), or estimated as the proportion of sensing instances in which the channel is correctly identified as idle, corresponding to the number of true negatives. Similarly, the interference rate to the primary user is estimated as the proportion of instances where the PU signal was present but not detected, corresponding to the number of false negatives. These values are derived from the confusion matrix provided earlier, yielding(47)$$\begin{array}{cc} \mathrm{Throughput} & =\frac{T_{N}}{N}=\frac{125}{200}=0.625, \end{array}$$(48)$$\begin{array}{cc} \mathrm{Interference}\;\mathrm{Rate} & =\frac{F_{N}}{N}=\frac{8}{200}=0.04. \end{array}$$

The SU is granted reliable transmission opportunities in 62.5% of the sensing intervals, enabling efficient spectrum reuse without compromising regulatory constraints. Meanwhile, interference with PU operations occurs in only 4.0% of the intervals, reflecting the sensing algorithm’s effectiveness in minimizing harmful overlap. This balance is essential in dynamic spectrum access scenarios where both spectrum efficiency and PU protection must be simultaneously addressed.

### 5.2. Geolocation Scenario and Metric Computations

This subsection illustrates spatial performance metrics using a synthetic example that simulates the behavior of a spectrum hole geolocation algorithm. Assume that the true locations of 10 spectrum holes are compared with the estimated ones, yielding the following absolute position errors (in meters):[4.6,5.9,6.7,3.8,8.2,7.3,6.1,9.0,5.5,4.9].

These values are used to compute point-wise and distributional metrics of geolocation accuracy, as follows.

#### 5.2.1. RMSE and MAE

The root mean square error and mean absolute error are given by(49)$$\begin{array}{cc} \mathrm{RMSE} & =\sqrt{\frac{1}{10}\sum_{i=1}^{10}e_{i}^{2}}\approx 6.42\;m, \end{array}$$(50)$$\begin{array}{cc} \mathrm{MAE} & =\frac{1}{10}\sum_{i=1}^{10}\left|e_{i}\right|\approx 6.2\;m. \end{array}$$

The geolocation algorithm exhibits moderate spatial error, with an average deviation of 6.2 m (MAE) and a root mean square deviation of 6.42 m. The small difference between RMSE and MAE suggests that the error distribution is relatively uniform, with no significant outliers.

#### 5.2.2. Geolocation Coverage and CDF

Let a fixed spatial error threshold of 8 m be used to evaluate spatial reliability. Among the 10 estimates, 9 fall within this margin. Then, it follows that(51)$$\begin{array}{cc} \mathrm{Coverage}\;(\mathrm{within}\;8\;m) & =\frac{9}{10}=90\%, \end{array}$$(52)$$\begin{array}{cc} \mathrm{CDF}@6.5\;m & =\frac{7}{10}=70\%. \end{array}$$

The geolocation system achieves 90% coverage within an 8-m tolerance, meaning that 9 out of 10 estimated locations lie sufficiently close to their respective ground truth values. Furthermore, 70% of the errors are smaller than 6.5 m, indicating that the algorithm consistently provides high spatial accuracy.

#### 5.2.3. Confidence Region

The confidence region is approximated based on the spatial dispersion of the estimated positions around their true values. For illustration, assume the radius of the 95% confidence ellipse is approximately 9.2 m. This means that, with 95% confidence, the true location of the spectrum hole lies within a 9.2-m radius around the estimated point. This probabilistic bound offers spatial guarantees on location uncertainty, which can guide regulatory protection zones and SU exclusion areas.

#### 5.2.4. SHGDR and IPR

Suppose a 10 × 10 grid is evaluated for spatial labeling (idle or occupied), and the spectrum hole geolocation classifier correctly labels 91 of the 100 spatial regions. Additionally, among 25 SU transmission zones, 3 unintentionally overlap with PU-protected areas. The spectrum hole geolocation detection rate and the interference-to-primary ratio are respectively given by(53)$$\begin{array}{cc} \mathrm{SHGDR} & =\frac{91}{100}=0.91, \end{array}$$(54)$$\begin{array}{cc} \mathrm{IPR} & =\frac{3}{25}=0.12. \end{array}$$

The spectrum hole geolocation detection rate of 91% reflects high spatial classification accuracy. The interference-to-primary ratio of 12% indicates that the geolocation errors led to SU-PU overlap in only a small fraction of the active transmission zones, suggesting that the system maintains good spatial separation between users while enabling spatial reuse.

## 6. Conclusions

This paper has presented a comprehensive survey of performance metrics for evaluating spectrum sensing and spectrum hole geolocation in the context of dynamic spectrum access (DSA) in cognitive radio networks. Grounded in the principles of statistical decision theory, the discussion covered a broad range of metrics, spanning binary hypothesis testing, signal detection, and spatial geolocation, tailored to both algorithmic evaluation and system-level considerations.

For spectrum sensing, fundamental metrics such as probability of detection, probability of false alarm, and probability of missed detection were reviewed, along with aggregate indicators like accuracy, balanced accuracy, and the F1 score. Trade-off visualizations using ROC and DET curves were also highlighted as essential tools for threshold tuning and classifier comparison. In addition, throughput and interference metrics were discussed to emphasize the practical implications of sensing errors on both secondary and primary users.

In the geolocation domain, we examined metrics that quantify spatial accuracy and reliability, including RMSE, MAE, geolocation coverage, confidence regions, and the spectrum hole geolocation detection rate. These were complemented by interference-aware metrics, such as the interference-to-primary ratio, which directly relate geolocation precision to regulatory compliance and coexistence constraints.

The taxonomy table and comparative trade-off discussion provided a structured view of the metric landscape, clarifying the roles, strengths, and limitations of each performance indicator. A case study was also proposed to illustrate the combined use of sensing and geolocation metrics in a simulated DSA scenario, presenting numerical examples and interpretations of the associated metrics.

Overall, this survey serves as a reference for researchers and system designers seeking to evaluate and optimize spectrum sensing and spectrum hole geolocation strategies. Future work may extend this framework to include metrics for adversarial environments and learning-based detection systems, which are increasingly relevant in the evolving landscape of 6G and beyond.

## Figures and Tables

**Figure 1 sensors-25-03770-f001:**
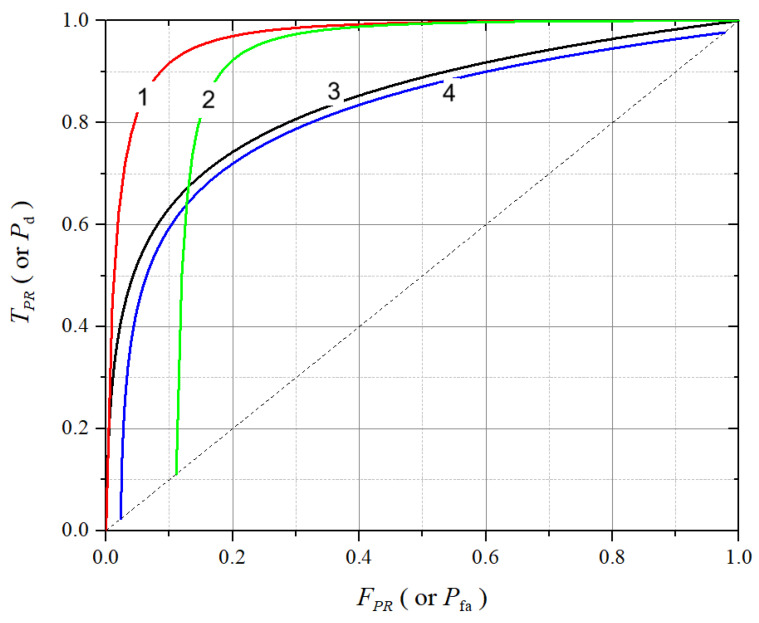
Examples of ROC curves.

**Figure 2 sensors-25-03770-f002:**
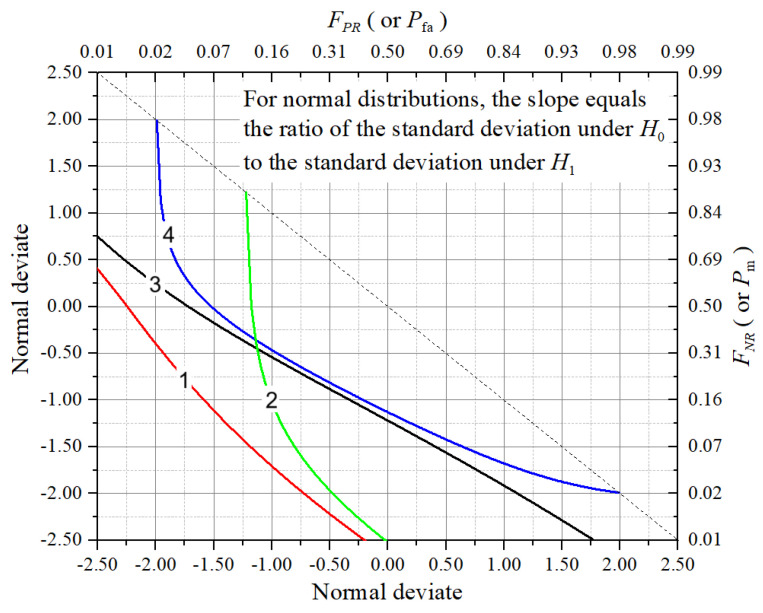
Examples of DET curves.

**Table 1 sensors-25-03770-t001:** Confusion matrix.

	Predicted Busy	Predicted Idle
Actual busy	True positives ($T_{P}$)	False negatives ($F_{N}$)
Actual idle	False positives ($F_{P}$)	True negatives ($T_{N}$)

**Table 2 sensors-25-03770-t002:** Comparison between ROC and DET curves.

Feature	ROC Curve	DET Curve
Axes	$P_{d}$ vs. $P_{\mathrm{fa}}$	$P_{m}$ vs. $P_{\mathrm{fa}}$
Scale	Linear	Normal deviate (probit)
Shape	Convex, top-left	Near-linear for Gaussian errors
Interpretability	Detection vs. false alarms	Error trade-off visualization
Use case	General performance overview	Emphasis on error balancing

**Table 3 sensors-25-03770-t003:** Taxonomy of performance metrics in spectrum sensing and spectrum hole geolocation.

Metric	Domain	Type	Strength	Limitation	Use Case
$P_{d}$	Sensing	Accuracy	Protects PUs By Avoiding Interference	May Raise $P_{\mathrm{fa}}$ If Too Sensitive	Evaluate Detection Capability
$P_{\mathrm{fa}}$	Sensing	False Positive Rate	Highlights Over-Cautious Sensing	Reduces SU Throughput	Threshold Tuning for SU Access
$P_{m}$	Sensing	False Negative Rate	Assesses Interference Risk	Inversely Related To $P_{d}$	Estimate Protection To PUs
Accuracy	Both	Overall	Easy To Compute And Interpret	Misleading With Imbalance	General Performance Assessment
Balanced Accuracy	Both	Imbalance-Resistant	Accounts For Skewed Class Proportions	Less Informative About Error Types	Used When PUs Active Infrequently
F1 Score	Both	Error Balance	Merges $P_{d}$ And Precision	Ignores TNs; Less Intuitive	Balanced Performance Metric
ROC/AUC	Both	Threshold-Free	Captures Performance Trade-Offs	May Require Probabilistic Output	Compare Detection Algorithms
DET Curve	Both	Error Trade-Off	Suits Gaussian Error Patterns	Less Widely Used; Needs Scale Transform	Visualize $P_{\mathrm{fa}}$ vs. $P_{m}$ Trade-Off
Throughput (SU)	Sensing	System Utility	Reflects Real-World Efficiency	Not Per-User; Ignores Fairness	Optimize SU Access Strategies
Geolocation Accuracy	Geolocation	Spatial Error	Intuitive Distance Error Metric	Affected By Outliers	Basic Localization Validation
RMSE/MAE	Geolocation	Aggregate Error	Capture Average Geolocation Error	Mask Local Variations	Statistical Quality Reports
Geolocation Coverage	Geolocation	Area Coverage	Indicates Where Estimates Are Precise	Threshold-Dependent	Define Usable Spatial Regions
Interference-To-PU Ratio	Geolocation	Interference Risk	Reflects Spatial Safety Margin	Requires Physical Modeling	Ensure QoS For Licensed Users

**Table 4 sensors-25-03770-t004:** Confusion matrix for the numerical examples.

	Predicted Busy	Predicted Idle
Actual busy	$T_{P}=52$	$F_{N}=8$
Actual idle	$F_{P}=15$	$T_{N}=125$

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
