# Peer review of "A Survey of Performance Metrics for Spectrum Sensing and Spectrum Hole Geolocation for Wireless Spectrum Access"

_sensors, 2025, doi:10.3390/s25123770_

Round 1
Reviewer 1 Report
Comments and Suggestions for Authors
This paper presents a survey of performance metrics applicable to spectrum sensing and spectrum hole geolocation within the context of dynamic spectrum access (DSA) in cognitive radio networks.
A key contribution of this work is to provide researchers and practitioners with a comprehensive set of evaluation tools, extending well beyond the applicability of the conventional probabilities of detection and false alarm.
Author Response
Comment 1: A key contribution of this work is to provide researchers and practitioners with a comprehensive set of evaluation tools, extending well beyond the applicability of the conventional probabilities of detection and false alarm.
Response 1: Many thanks for reviewing my manuscript and for the positive feedback.
Reviewer 2 Report
Comments and Suggestions for Authors
This survey is related to performance metrics for spectrum sensing in dynamic spectrum access (DSA) cognitive radio networks. The survey is well structured and gives full picture of different metrics and provides guidelines for their practical application in different scenarios including detection of false alarm and spectrum holes locations. Abbreviations sections is very informative because many abbreviations are used in the survey.
Some minor comments could be made in order to make further improvement of the paper.
Keywords: it seems reasonable to add “Performance metrics”
Introduction section: More fresh references could be added in the field of spectrum sensing dated 2023 – 2025 (several articles devoted to this topic have been published in Sensors recently)
Section 2 (Statistical Foundations): it is also recommended to add some recent literature references in this research area, because spectrum sensing theory has been actively developing in recent years. Application of well-known theory for spectrum sensing in Cognitive radio could be mentioned in this section.
More graphical illustrations of metrics are recommended in Sections 3 and Section 5 (like e.g. figures 1 and 2) – it will be more illustrative and interesting for the readers.
For Table 3 maybe references on literature are needed because it summarizes known metric Types based on literature.
Conclusion section: It is recommended to reflect the connection with Section 5 for a more complete picture of the survey and to support it with experimental analysis made in Section 5.
Thank You for the opportunity to review
Such an interesting and useful paper.
Beat Regards,
The reviewer
Author Response
Comment 1: Keywords: it seems reasonable to add “Performance metrics”
Response 1: Thanks for the suggestion. I have added the keyword, as suggested.
Comment 2: Introduction section: More fresh references could be added in the field of spectrum sensing dated 2023 – 2025 (several articles devoted to this topic have been published in Sensors recently).
Response 2: Many thanks for the suggestion. A new Section 1.1 - Related Work has been added and new references included. A total of five new references were added in the revised manuscript, mainly those that are highly cited in the literature.
Comment 3: Section 2 (Statistical Foundations): it is also recommended to add some recent literature references in this research area, because spectrum sensing theory has been actively developing in recent years. Application of well-known theory for spectrum sensing in Cognitive radio could be mentioned in this section.
Response 3: Many thanks for the suggestion. As mentioned in Response 2, five new references were added in the revised manuscript to cover recent and highly-cited works on spectrum sensing. I believe that these references can cover extensively the theory of spectrum sensing in general. On the other hand, I was not capable of finding solid recent works covering the statistical decision theory applied specifically to spectrum sensing, beyond the classical references well recognized in the literature and already included in the manuscript. Nonetheless, two more references were added at the beginning of Section 2 - Statistical Foundations for Spectrum Sensing, to increase the rol of opportunities for the reader to seek for more information in this area.
Comment 4: More graphical illustrations of metrics are recommended in Sections 3 and Section 5 (like e.g. figures 1 and 2) – it will be more illustrative and interesting for the readers.
Response 4: Many thanks for the suggestion. Among the metrics covered, only the ROC and DET curves are simple graphical tools for evaluating spectrum sensing performance, hence the decision for including only them as examples. In fact, any other metric could be, in principle, inserted in a graph. For example, the spectrum hole geolocation detection rate can be evaluated as a function of important system parameters, like the SNR, the number of secondary users, and the number of samples (please, see reference [39] of the revised manuscript). Other graphical metrics like the confidence region or the CDF of geolocation coverage could be inserted as well. However, the inclusion of such graphs, would need input data that would demand a minimal explanation (not short, unfortunately) about system models and parameters, and how these data were generated, eventually deviating from the purpose of presenting the metrics and giving synthetic numerical examples of their usage and interpretation (which is made in Section 5).
Comment 5: For Table 3 maybe references on literature are needed because it summarizes known metric Types based on literature.
Response 5: Many thanks for the comment. I have modified the text calling Table 3 to highlight that the table has been constructed to summarize the main characteristics of the metrics previously covered throughout the text. I did not add references in the table because only a few metrics are typically covered in the literature (the probabilities of detection and false alarm) - hence the contribution of the manuscript in suggesting a larger set of metrics that could be used to assess spectrum sensing performance well beyond the classical probabilities of detection and false alarm.
Comment 6: Conclusion section: It is recommended to reflect the connection with Section 5 for a more complete picture of the survey and to support it with experimental analysis made in Section 5.
Response 6: Thanks for the suggestion. A complementary phrase has been added to the Conclusions section in the revised manuscript as an attempt to connect to Section 5.
Comment 7: Thank You for the opportunity to review such an interesting and useful paper.
Response 7: Many thanks for reviewing my manuscript and for the very positive feedback.
Reviewer 3 Report
Comments and Suggestions for Authors
In this article, the authors have presented a survey of performance metrics applicable to spectrum sensing and spectrum hole geolocation within the context of dynamic spectrum access (DSA) in cognitive radio networks. Further, highlights trade-offs among metrics and provides guidelines for their practical application. The presented research work is very interesting; however, I do not recommend its acceptance in its present form. It needs revision as follows.
- In this survey/review article regarding spectrum sensing for cognitive radio networks, the author must clarify the contributions particularly in bullet form. I think, the author must tabulate all the reported survey/review article with their contribution and then clarify the novelty of this manuscript.
- The presented spectrum sensing performance metrics are applicable on all spectrum sensing techniques such as Energy detection, Match Filter, cyclostationary etc..
- In the spectrum Sensing technique for cognitive radio, the threshold selection is a very important concern which is missing in whole manuscript.
- In cognitive radio networks, cooperative spectrum sensing play very significant role, for which selection rule at fusion centre with perfect and imperfect reporting channel are also very important concerns. I think, the author must discuss these parameters also. I recommend some google scholar links for several articles (author may select best, and reputed among them).
[R1] 10.1109/JSEN.2022.3142197
Author Response
Comment 1: In this survey/review article regarding spectrum sensing for cognitive radio networks, the author must clarify the contributions particularly in bullet form. I think, the author must tabulate all the reported survey/review article with their contribution and then clarify the novelty of this manuscript.
Response 1: Thank you for this pertinent suggestion. In response, I have restructured the Introduction section of the manuscript. Specifically, I have added two new subsections, Section 1.1 (Related Work) and Section 1.2 (Contributions and Organization of the Paper), to clearly distinguish the manuscript’s novelty. Section 1.1 now summarizes key existing survey/review/tutorial papers and their primary contributions. This analysis helps highlight the specific gap addressed by the present work. Section 1.2 then explicitly highlight the main contribution of my manuscript, emphasizing that, unlike prior reviews, this work is solely dedicated to performance metrics for spectrum sensing, with a comprehensive treatment that extends well beyond the conventional metrics of detection and false alarm probabilities.
Comment 2: The presented spectrum sensing performance metrics are applicable on all spectrum sensing techniques such as Energy detection, Match Filter, cyclostationary etc?
Response 2: Thank you for raising this important point. Yes, the performance metrics discussed in this manuscript are designed to be technique-agnostic, meaning they are applicable to all spectrum sensing techniques, including energy detection, matched filtering, cyclostationary detection, and others. Furthermore, several of the addressed metrics extend beyond the traditional evaluation of individual detectors. In particular, the manuscript also discusses metrics tailored to spectrum hole geolocation, thereby enabling the performance assessment of spectrum sensing in the spatial domain, a dimension often overlooked in traditional reviews.
Comment 3: In the spectrum Sensing technique for cognitive radio, the threshold selection is a very important concern which is missing in whole manuscript.
Response 3: I appreciate your observation. Indeed, the selection of the detection threshold is a fundamental concern in spectrum sensing, and it is addressed at multiple points throughout the manuscript. To clarify this, I now explicitly emphasize these references in the revised version: i) after Equation (5) in the context of the Neyman-Pearson criterion; ii) after Equation (8) in the context of Bayesian approach to spectrum sensing; iii) in the last paragraph of Section 3.9 and at the beginning of Section 3.10, in the context of ROC curves; iv) in the fourth bullet of Section 3.11, and in the first paragraph after Fig. 2, in regard to DET curves; v) in the middle of Section 3.14, where the negative likelihood ratio is addressed; vi) in the last paragraph of Section 3.16, in the context of the logarithmic loss; vii) in Section 3.17, where the p-value is explored; viii) in Section 3.18, regarding the detection time; ix) in the first paragraph of Section 4.5, where the interference to primary ratio in geolocation is addressed; and in the Conclusions.
Comment 4: In cognitive radio networks, cooperative spectrum sensing play very significant role, for which selection rule at fusion centre with perfect and imperfect reporting channel are also very important concerns. I think, the author must discuss these parameters also. I recommend some google scholar links for several articles (author may select best, and reputed among them). [R1] 10.1109/JSEN.2022.3142197
Response 4: Thank you for highlighting the role of cooperative spectrum sensing and the related issues of fusion rules under different reporting channel conditions. While the primary focus of the manuscript is not on the mechanisms of spectrum sensing themselves, including cooperative or non-cooperative approaches, it aims to provide a set of performance metrics that are universally applicable, regardless of the specific sensing architecture. These metrics can indeed be used to evaluate the effectiveness of cooperative sensing schemes, including those involving imperfect reporting channels. Nonetheless, I acknowledge the value of your suggestion and have now cited the recommended reference [R1] (10.1109/JSEN.2022.3142197) in Section 3.10, within the discussion on ROC and DET curves.
Round 2
Reviewer 3 Report
Comments and Suggestions for Authors
In the revised manuscript, the authors have incorporated almost all the comments and suggestions raised by the reviewers. Therefore, I would like to recommend the acceptance of this revised manuscript for publication.